# Human Activity Recognition of Individuals with Lower Limb Amputation in Free-Living Conditions: A Pilot Study

**DOI:** 10.3390/s21248377

**Published:** 2021-12-15

**Authors:** Alexander Jamieson, Laura Murray, Lina Stankovic, Vladimir Stankovic, Arjan Buis

**Affiliations:** 1Wolfson Centre, Department of Biomedical Engineering, University of Strathclyde, Glasgow G4 0NW, UK; alexander.jamieson@strath.ac.uk (A.J.); laura.murray.100@strath.ac.uk (L.M.); 2Department of Electronic and Electrical Engineering, University of Strathclyde, Glasgow G1 1XW, UK; lina.stankovic@strath.ac.uk (L.S.); vladimir.stankovic@strath.ac.uk (V.S.)

**Keywords:** human activity recognition, lower limb amputation, physical activity, machine learning, lower limb prosthetics

## Abstract

This pilot study aimed to investigate the implementation of supervised classifiers and a neural network for the recognition of activities carried out by Individuals with Lower Limb Amputation (ILLAs), as well as individuals without gait impairment, in free living conditions. Eight individuals with no gait impairments and four ILLAs wore a thigh-based accelerometer and walked on an improvised route in the vicinity of their homes across a variety of terrains. Various machine learning classifiers were trained and tested for recognition of walking activities. Additional investigations were made regarding the detail of the activity label versus classifier accuracy and whether the classifiers were capable of being trained exclusively on non-impaired individuals’ data and could recognize physical activities carried out by ILLAs. At a basic level of label detail, Support Vector Machines (SVM) and Long-Short Term Memory (LSTM) networks were able to acquire 77–78% mean classification accuracy, which fell with increased label detail. Classifiers trained on individuals without gait impairment could not recognize activities carried out by ILLAs. This investigation presents the groundwork for a HAR system capable of recognizing a variety of walking activities, both for individuals with no gait impairments and ILLAs.

## 1. Introduction

Following an amputation procedure, Individuals with Lower Limb Amputation (ILLAs) have been shown to be less physically active than individuals without limb loss [1,2], and there has been extensive research into the physical and socio-economical barriers that prevent ILLAs from performing physical activity, such barriers include existing co-morbidities such as diabetes or chronic obstructive pulmonary disease and a lack of resources or group support [3,4,5]. Maintaining sufficient levels of physical activity is vital to ILLA’s physical and mental well-being: physical activity has evidently improved heart and lung functionality and reduced the effects of chronic lower back pain [4,6]. Physical activity has further given improved perceptions of the individual’s quality of life, self-esteem, and body image [4,7]. Therefore, it is imperative that healthcare professionals can monitor and evaluate the physical activity of their clients. Modern day healthcare of ILLAs is limited by a lack of efficient implementation of activity monitoring systems, as evidenced in a recent literature review [8], and is worthy of further research.

By performing automatic recognition and classification of different activities, it is possible to establish the basis of a reliable and low-cost activity monitoring system, in which healthcare professionals that specialise in the care and wellbeing of ILLAs (e.g., physiotherapists) can track how their patients remain active over long periods of time and in free-living conditions. Free-living Human Activity Recognition (HAR) studies are not uncommon, see e.g., [9,10,11], and there are publicly available datasets of individuals in free-living [12]. However, there are currently very few studies which have carried out a HAR exercise in free-living conditions with an ILLA population [13]. One of the few examples are the works of Zhang et al. [14,15]; however, their sensor configuration utilised RGB data from a camera as one of the inputs for terrain recognition process, which can have serious ethical and cost considerations if applied in a clinical monitoring context [16]. Additionally, while locomotion activities are very commonly analysed in ILLA-focused HAR studies, these are typically limited only to walking on level ground, ramps, and stairs [13]. In free living conditions, one would be far more likely to traverse hills instead of ramps, and it is worth highlighting the differences between them. Ramps are mandmade structures that typically exhibit a fixed incline over a short distance, while hills have variable incline over longer distances. Naturally, the latter can be more challenging for adequate machine learning recognition due to their intrinsic variations.

The primary objective of this pilot study was to investigate the implementation of supervised classifiers and a neural network for the recognition of activities carried out by Individuals with Lower Limb Amputation (ILLAs), as well as healthy individuals with no gait impairment, in free living conditions. Additionally, to the researchers’ knowledge, this is one of the first papers that have investigated the potential for terrain “resolution” in HAR for an ILLA population. The investigations carried out in this paper are divided into three experiments: The first step takes an assortment of classifiers and a deep learning network and uses hyperparameter tuning to determine which classifiers were the most suited for recognizing walking activities. Subsequently, the best classifiers from the first experiment were then used in the second step to investigate how the terrain “resolution” impacts on classifier performance. Finally, in the third step, there is an investigation as to whether the data trained on a group of subjects (either ILLAs or healthy individuals with no gait impairment) could recognize activity carried out by an ILLA.

## 2. Materials and Methods

### 2.1. Data Collection and Pre-Processing

Ethical approval for the pilot study was granted by the University of Strathclyde’s University Ethics Committee prior to conducting the investigation. Participants were recruited between September and December 2020 via social media posters and contacting people who had consented to follow-up participations from previous investigations. Due to lack of recruitment numbers (primarily caused by the Coronavirus), participants were selected based on convenience sampling. All participants had to be at least 18 years of age, be comfortable performing moderate vigorous activity, and not be at risk of life-threatening conditions if infected with the coronavirus. The ILLA volunteers were further required to be able to ambulate with a prosthesis without the use of walking aids and without any comorbidities or musculoskeletal conditions which could potentially impact their ability to ambulate for sustained periods of time. As the study was primarily targeted at monitoring the activity of ILLAs, many of which will have poor levels of mobility, participants were explicitly asked not to carry out vigorous activities such as jogging or running.

Recruited participants were instructed to carry out approximately 140 min of walking over the duration of a week (expecting a minimum of 20 min of walking per day) in the local vicinity of their homes while recording inertial data via a thigh-worn accelerometer. The accelerometer is an ActivPAL, presented in Figure 1 (PAL Technologies, Glasgow, UK). The ActivPAL is 4 × 2 cm, weighing approximately 0.01kg and is characterized with an operating frequency of 20Hz. The ActivPAL’s accuracy has been validated in terms of its ability to measure levels of activity intensity (including being able to distinguish sitting and standing behaviour) as well as accurately measure the step count with an ILLA population [17,18,19]. Ground truth validation of the activities were acquired by instructing the participant to additionally wear a chest-mounted camera and record elevation data via a GPS recording application (Strava, Strava Inc., CA, USA) operating on an iPhone 6 (Apple Inc., CA, USA) smartphone provided by the researchers. An example of the recording set-up is shown in Figure 2. Activities were annotated using the Visual object Tagging Tool (VoTT) (Microsoft, WA, USA), using the GPS co-ordinates from the Strava application to aid with labelling uphill and downhill movement; this is elaborated further in Section 2.2. All signal processing and machine learning was handled in a Matlab environment (Matlab 2020b, Mathworks, MA, USA). The accelerometer data were digitally filtered by a linear bandpass Butterworth filter with a preserved frequency range of 0.4 Hz to 3 Hz and order of 3 to remove the gravitational constant and high frequency muscle artifact movements from the raw signal [20,21]. The data were segmented into chunks of 2 s long (or 40 sample width) windows with no overlap, a two second window length would be adequate to capture at least one step, and the choice of no overlap in the sampling window was arbitrary due to the choice being non-consequential compared to other design choices in the machine learning process [22]. Longer sampling windows decreased the number of samples available for training, which was not ideal given the large imbalance of classes in the dataset, see Section 2.3 and Table 1. A list of 243 statistical, frequency and wavelet domain features (refer to Section A.1) were calculated for each segment of data and are standardized to have zero mean and a standard deviation of one. To give an abridged justification of feature inclusion, the list of features in this study were chosen as a combination of their widespread inclusion in similar HAR papers that cover exploratory data analysis (e.g., [13,23,24], or through analysis of the fundamental differences in acceleration between different types of activities. For example, to distinguish terrains with accelerometer data, a change in the ground reaction force when traversing different types of terrains (as evidenced in [25]) will result in a change of the accelerometer waveform. This will have impact in the wavelet domain (a domain that considers both time and frequency), and can therefore be quantifiable with wavelet features. The use of wavelet features for the distinguishment of anatomically similar activities has been further supported by recent literature [26].

### 2.2. Annotation of Slopes

To identify uphill and downhill segments of a walk, GPS data from Strava was utilised. The angle of slope being traversed as a function of time θ(t) was calculated by approximating the slope as a straight line and then calculating the inverse tangent of the elevation change (ΔY(t)) divided by the distance change (ΔX(t)). ΔY was obtained from the Strava elevation readings, while ΔX was calculated using WGS84 ellipsoid distance of latitude and longitude co-ordinates also taken from Strava’s GPS data. Elevation data in Strava recordings are generated based on a combination of barometer readings directly extracted from the operating smartphone’s sensor and, where applicable, a basemap of elevation data where points of elevation are known [27]. The iPhone 6 contains a commercial BMP280 barometer (Bosch, Gerlingen, Germany) which can measure changes in altitude with a precision of ±0.12 hPa (or ±1m equivalent), operating at a sample frequency of approximately 1 Hz [28]. A study of various commercial GPS devices found the Strava application running on an iPhone 6 device produced the lowest error in elevation and distance, and so was considered the best solution for low-cost GPS acquisition [29]. Since elevation readings were dependent on the accuracy of GPS readings, whose accuracy can fluctuate depending on the strength of the GPS signal among other factors [30], the angular values were given a smoothing average over an empirically determined 10 samples, which gave a reasonable trade-off between “stability” of the angular values plotted over time and the peak magnitude of the angle readings. A failsafe countermeasure was put in place where if there were more than 15 s between two successive Strava readings, the angle would not be calculated and all unclear movement within that timeframe would be given a “null” label, meaning no activities were annotated within this time period. An exemplary angle plot is demonstrated in the Figure 3. Due to the uncontrolled nature of the investigation, the absolute angle of slopes in terrain traversed by the participants were unknown prior to the investigation, making the accuracy of the slope readings not possible to validate. This should be acknowledged as a key limitation of the slope readings, and would require subsequent validation in future studies using an outdoor structure with fixed and known angle of slope.

As annotation of the video progressed, the annotator would periodically check for whenever the thresholds in the slope angle data were crossed. There were four key angular thresholds in the annotation process: a slope would be determined as uphill when the angle of the slope exceeded +2.9°, or −2.9° for downhill. This threshold was referenced from the American Disability Association’s minimum slope angle requirement for construction of a handrail in a public access building [31]. Due to the imprecision of elevation and ellipsoid distance readings, an “unconfident” threshold was also constructed between +1.45 to +2.9° and −1.45 to −2.9°. Any data that fell within this threshold would be given a null label, the exception being when a participant was traversing stairs. As it could not be confidently ascertained whether readings within this threshold were uphill, downhill, or flat, it would have led to numerous mislabelling instances; thus, readings within these ambiguity thresholds are discarded from the process.

### 2.3. Introducing *“Label Resolution”*

In this paper, annotated activities are initially given what is referred to as “full label resolution”, such that each sample is described with its terrain and the condition of the terrain. For example, walking along a flat concrete road is initially annotated as “Concrete, Flat”, while walking uphill on a grassy knoll is annotated as “Grass, Uphill”. The exception to this is stair traversals, which are referred to as “Upstairs” or “Downstairs”. However, at such a level of detail, this resulted in 24 unique activities (as seen in Table 1), thus will adversely impact on the machine learning system. Subsequently, samples undergo a process called “label consolidation”, where similar activities are given the same label. As a result, there are three levels of Label Resolution used in this study:Level 1: No Terrain Resolution. Labels only contain the condition of the traversed ground (ground is flat, ground is sloped, or is traversing stairs)Level 2: Hard/Soft Terrain Resolution. Terrains are resolved into whether the terrain involved is hard/soft. Hard terrains included concrete, stone, and gravel, while soft terrains included grass, sand and red ash (a type of clay pitch).Level 3: Full Resolution: All Labels use their full label resolution.

To further illustrate the consolidation process, a flowchart in Figure 4 has been created to show how the process operates for all activity labels that relate to traversing flat terrain. The process is identical for all uphill and downhill related activities.

In the first and third experiments, the annotated labels were assigned Level 1 terrain resolution, this meant that there were only five classes for potential classification: flat walking, uphill walking, downhill walking, upstairs movement, and downstairs movement. This was designed to give an idea of how the classifiers would perform at the simplest level of classification and is comparable to the labels used in other studies with ILLAs [13]. Due to the unstructured nature of the investigation, there was a large imbalance between labels; at Level 1, flat walking comprises approximately 70% of the total dataset, while stair data (up and down) both comprise only around 5% combined. In order to avoid having a highly biased classifier that favours prediction of the majority class [32], classes were rebalanced using Synthetic Minority Oversampling Technique (SMOTE) [33].

### 2.4. Experiment 1: Comparison of Supervised Classifiers

The objective of the first experiment was to identify the best classifier for supervised classification of the five walking activities (Level 1 resolution). Six off-the-shelf supervised classifiers were tested: Support Vector Machine (SVM), k Nearest Neighbours (kNN), Random Forest, Adaboost, Naïve-Bayes, Linear Discriminant Analysis (LDA) and were compared with an additional Long Short-Term Memory Network (LSTM). The classifiers were chosen based on their frequent appearance in HAR studies (see e.g., [34,35,36,37,38,39]). After heuristic experimentation with various dimensionality reduction techniques, it was found that application of Principal Component Analysis (PCA) with express preservation of components that explained 95% of the total systematic variance (approximately 65–70 components) was a suitable technique for dimensionality reduction for the off-the-shelf classifiers. The LSTM model constructed for this investigation was a Sequence-to-Label network demonstrated in Figure 5. Experimentation with feature selection indicated that the LSTM network performed best when using each axis of the raw accelerometer signal (post-filtering) as 3 separate channel inputs. Using guidelines set by Bengio [40] and some heuristic experimentation, the most suitable parameters defined for the network were to perform ADAM Optimization [41], set mini batch size to 256, set the initial learn rate to 0.005 with a decrease in the learn rate by a factor of 10 every 2 training epochs, and set the maximum number of epochs to 15. These parameters were kept consistent throughout all stages of analysis because they had minimal influence on training accuracy, and so this avoided cumbersome tuning times for the LSTM network. Raw training data is also balanced via the application of SMOTE.

The six tested supervised classifiers with embedded feature selection, and the LSTM network with raw data, underwent hyperparameter tuning to determine the best classifier. The list of eligible hyperparameters for tuning are listed in Table 2. For the SVM models, Kernel Scale variations and One vs. All training were excluded from tuning as they resulted in training times exceeding an hour per optimization iteration. For the supervised classifiers, the tuning process was performed with Random Search with 20 maximum iterations [42]; this excludes Naïve-Bayes and LDA classifiers which had less than 20 possible input combinations. The LSTM network, due to its long training times, used a simple grid-search approach with discrete values for the dropout factor (0.1 to 0.5) and the number of hidden units (175 to 350).

All classifiers were validated using 5-Fold cross validation of the entire dataset: a combination of all ILLA and non-impaired individuals’ data. Though 10 is generally the standard value of the K in cross validation in standard HAR papers, it was found that training times, particularly of the LSTM network, were very long (upwards of 10 min per iteration), and therefore was unsuitable for 10-fold. 5-Fold is generally considered to be the minimal acceptable level of K for machine learning validation [43]. Parameter tuning for the supervised classifiers used a nested cross validation approach [44]. For outer $k_{o}$ folds, the data is split into training and testing. The training data is further split into an inner sub-fold $k_{i}$ to tune hyperparameters via k-fold cross validation, with both the inner and outer loop values for k set to five. Thus, for each $k_{o}$ fold, a set of optimal parameters are obtained, and the classifier is trained with these parameters. The classifier is validated with the test-set data to get an accuracy, and the overall reported accuracy is the mean value of test accuracy across the $k_{o}$ folds. While LSTM used 5-folds in the outer cross validation, due to very long training times, the internal validation process for parameter tuning instead used a holdout of 25% of the training data.

### 2.5. Experiment 2: Label Terrain Resolution

In this experiment, the appropriate level of terrain resolution is determined. There was an expected trade-off between the resolution of detail in the activity and the corresponding classifier performance. This experiment aimed to validate this theory, and to show at what specific label of terrain resolution can an acceptable level of classifier accuracy be obtained. Using the best performing classifiers from experiment 1, the terrain resolution was increased in stages from level 1 to level 3 as per the criteria described in Section 2.3. Each level of resolution is validated using 5-fold cross validation of the entire dataset.

### 2.6. Experiment 3: Subject Cross-Validation

In the final experiment, the objective was to determine whether a classifier trained exclusively on either healthy or ILLA participants would be robust enough to detect physical activities in a single ILLA participant. Hence, this experiment uses Leave One Subject Out (LOSO) validation. There were two methods of training the classifier:Method 1: Healthy Participant Training. For the best performing classifiers, train data on the healthy participants, then use LOSO validation to test the classification accuracy for each of the ILLA participants individually.Method 2: Amputee Participant Training. All data from healthy participants are ignored. LOSO validation is again performed with the best classifiers, this time training the data on all-minus-one ILLA participants and testing on the remaining ILLA participant, repeating the process for each ILLA.

In this experiment, the label resolution is kept at Level 1.

## 3. Results

### 3.1. Participant Information

A total of eight healthy participants with no gait impairment and four participants with lower limb amputation were recruited for the study. From this point onwards, specific participants are referred to using a letter-number rule. For example, healthy participant #8 is referred to simply as “H8”, and amputee participant #2 is referred to as “A2”. All participants who consented to participate in the study were able to carry out their recording successfully, with zero dropouts. However, two of the recordings from participant H1, and one recording each from participants H4 and H5 had to be discarded from analysis due to improper camera wear or recording at night-time, making analysis of those recordings infeasible. The participants’ primary characteristics are listed in Table 3 and Table 4. The demographics match in terms of height and weight, however there were large gender biases in both demographics with the vast majority of participants being male. Additionally, there was a noticeable age bias between healthy and amputee demographics; most of the healthy volunteers were in their mid-20s, while the mean age of the ILLAs is approximately 50. While this could have had adverse impact on machine learning testing accuracies, for research purposes it would be useful to determine if training data on a younger healthy population could still result in the detection of activity for an older population with gait impairments. Despite the relatively small sample size of ILLA participants, there was some interesting variation in the amputation time with 2 long-term experienced ILLAs and two comparatively inexperienced ILLAs, and there was a bilateral amputee to provide a comparison point to unilateral amputees. Contrary to Scottish national statistics [45], all but one of the amputees had vascular-related issues as the primary cause of amputation, with the other three having traumatic-related causes.

### 3.2. Experiment 1: Classifier Optimization

The results of the first experiment are shown in Table 5, and from these the “best” classifiers were identified as the SVM and LSTM models. Both models were considered to be equally viable due to the high fold-dependent standard deviation in the LSTM model, while SVM had much more consistent recognition accuracy across each fold. The best parameters indicate the hyperparameters that were present in the highest scoring fold of the outer cross validation.

### 3.3. Experiment 2: Label Terrain Resolution

All subsequent experiments used SVM and LSTM models with their best parameters from Table 5. The mean classification accuracy results from varying terrain resolution can be seen in Table 6, which is accompanied with F1 scores for each activity in Table 7, Table 8 and Table 9. In the base terrain resolution, flat and stair data had notably greater F1 scores compared to hill data, which is to be expected due to stair movement having significantly different kinematic behaviour from standard walking [46]. Uphill movement typically had less confusion than downhill movement, which is supported by Kimel-Naor et al [47], whose findings suggest that downhill movement and flat movement are often very similar due to minimal required changes in gait kinematics for downhill movement. The mean accuracy of both classifiers decreases as terrain resolution (and in turn, the number of activities) increase. While the drop-off in mean classification accuracy for the LSTM model was less severe than the SVM model, the F1 scores at each stage of terrain resolution (Table 7, Table 8 and Table 9) reveal that that many of the individual classes for Level 2 and 3 have poor F1 scores, and the SVM model had larger F1 scores in many class-by-class comparisons. The confusion matrices for the LSTM and SVM models at each level of terrain resolution are contained in Appendix B. By viewing the confusion matrices of the LSTM model (Refer to Figure A4, Figure A5 and Figure A6 in Appendix B), the higher accuracies are a result of having high recall for the majority class (“Flat”, “Hard, Flat” or “Concrete, Flat”) indicating that despite balancing the classes with SMOTE, the LSTM model has high model complexity and was overfitting [48]. The SVM models (Figure A1, Figure A2 and Figure A3), while having relatively worse performance, had less evidence of underfitting and better precision in the minority classes.

### 3.4. Experiment 3: Subject Cross-Validation

Table 10 and Table 11 present the results of training classifiers by an ILLA or healthy population and testing on one ILLA, using Level 1 or “no terrain” resolution. Clearly, this experiment was unsuccessful; while SVM accuracies are mediocre, the LSTM models failed completely, for some participants having a recognition accuracy lower than random guess. Despite the significantly smaller size of training dataset available, the mean classification performance of the SVM when trained by healthy participants was lower than the performance when the SVM was trained by other ILLAs, indicating that training a classifier via individuals with unimpaired gait are not suitable for classification of activities carried out by ILLAs. A significant confounding factor in this study was the fact that the ILLAs walked over considerably more variations in terrain than the healthy individuals, and even within the ILLA group, some ILLA walked on certain terrains that other ILLAs rarely or never walked over. For instance, participant A1 walked over stony terrain, with no instances of sandy terrain, while the opposite was true for participant A2. Participant A4 being a bilateral amputee, while the other three were unilateral, may have caused further confusion in the models. The LSTM model had wildly inconsistent inter-participant accuracy and high standard deviations, but given the much smaller size of training dataset, particularly when trained with only ILLA participants, this was anticipated. The confusion matrices for each participant (by Classifier and by training method) are located in Appendix B. Despite the overall poor findings, a positive perspective of these results is that the years of amputation experienced by the amputee (refer to Table 4) do not appear to influence the validation results, judging by the comparable performances of participant A1 (amputation time 3 years) to participants A2 and A3 (amputation time 32 and 33 years respectively). Though, due to the minimal sample size, this is not a generalizable result.

## 4. Discussion

### 4.1. Main Findings and Interpretations

By analysing walking activity data carried out by ILLAs and healthy individuals in free-living settings with supervised classifiers, this research was able to achieve satisfactory results, but with significant room for improvement. The best suited classifiers for the supervised approach were discovered to be an SVM which calculated 243 features in the time, frequency, and wavelet domains, with compressed dimensionality via PCA to 68 principal components, and an LSTM Network which required only the raw sequential triaxial accelerometer data. The LSTM model generally had superior mean classification accuracy but required much longer training times and had perceptible overfitting on the majority class. Given these significant drawbacks, the SVM was overall the more robust classifier and, if based purely on these results, would be the most suitable classifier in a clinical-based activity monitoring system. The LSTM’s design is based on a simple preconfigured model available from Matlab’s Deep Learning designer application. In cutting-edge deep learning HAR experimentation, LSTM models tend to exhibit complicated designs. Zhao et al. [49] for instance implemented a bidirectional LSTM which learns time dependencies in both directions of the input signal. Using this model, Zhao et al. [49] was able to recognize the five activities listed in Level 1 terrain resolution of the Opportunity dataset [50], acquiring an average of 94% in F1 scores across the five activities for a 70/30% split of training and test subject data. Sun et al. [51] meanwhile used a combination of CNN and LSTM architecture. The CNN component performed feature extraction from convolution of the raw input data, from which the LSTM recognizes the time-dependencies of the CNN-forged features. They performed classification through an Extreme Learning Machine—an alternative to the SoftMax classifier [52], and were able to achieve strong recognition accuracies for high level activities. Implementation of the described LSTM variants from [49,50,51] may have resulted in improved classification results in this study. Technical limitations of the deep learning networks arose as a result of Matlab’s lack of official support for deep learning classifiers; for example, Extreme Learning Machines are not supported in their deep learning classification app and require downloading unofficial community-supported toolboxes in place. Future implementations of the project will utilize Keras in a Python™ (Python Software Foundation, DE, USA) Integrated Development Environment (IDE) for more refined control of deep learning parameters as well as for accessing a wider range of implementable deep learning libraries [53].

From the terrain-resolution experiments (experiment 2), there is some potential to expand the typical walking activities captured in a HAR investigation to include hard and soft terrains (label resolution: Level 2) as additional outcomes. While “soft, flat” had acceptable F1 scores, “soft, uphill” and “soft, downhill” did not, this is likely due to having much smaller quantities of these activities compared to “soft, flat” (1604 for "soft, flat" vs. 109 and 163 for "soft, uphill" and "soft, downhill" respectively) and the hard terrain activities (which total around 24,000 samples). While the inclusion of SMOTE can help increase performance in these minority classes and prevent overfitting [54], if the initial training data size is too small, the SMOTE process will still be unable to successfully capture the full range of potential feature data that minority class can exhibit because it can only create new samples within a nearest neighbour proximity to that existing data, which can be especially problematic if the data is noisy [55]. Balancing the dataset in the future may require greater degrees of undersampling in the majority class to provide a better initial balance to the data prior to SMOTE. Unfortunately, Level 3 label resolution appears to be impractical to detect accurately, at least within the scope of the data available for the investigation. There would likely require significantly more data from the minority classes and likely require more additional sensors. For example, a second ActivPAL attached to the other leg could use inter-sensor axis correlation coefficients to detect the presence of laterally uneven ground. The presence of “camber movement” in particular may be impossible to detect with consistent accuracies due to its high similarity with “concrete, flat” movement coupled with variations in road designs (for example, having different degrees of slope).

The subject cross-validation experiments (experiment 3) were regrettably completely unsuccessful, even at the most basic level of terrain resolution. As briefly mentioned in the results, this likely was due to the ILLAs carrying out different walking activities, both from the healthy individual group and from each other. Future investigations would need to acquire considerably larger numbers of ILLA volunteers with a diverse range of types of amputation, and work towards avoiding the gender and age biases shown in this dataset which may have had further influence on the results. As it stands, the clinical implications the final experiment is that a classifier for the detection of walking activities in ILLAs cannot be reliably trained on activity data from a population with non-impaired gait, and by comparing the LOSO validation of the bilateral amputee (participant A4) to the others, this experiment also indicates that different classifiers may need to be trained for different types of amputation.

### 4.2. Comparisons with Other Studies

The recognition accuracies in this investigation were far from ideal: each of the lab-based studies identified in [13] that used only an IMU sensor for data collection had comparable or superior recognition accuracies [56,57,58,59,60]. This pilot study is however does have comparable performances with other free-living HAR investigations that use a healthy gait-unimpaired population. Gyllensten and Bonomi. [61] found that activity data trained in laboratory conditions dropped from an average recognition accuracy of 95.4% to 75.6% with an SVM. Their standard deviations for the SVM model were also considerably higher (±10.4% compared to 0.54% in this study), though this difference may be explained by measuring different activities and using Leave-One-Subject-Out Cross Validation. Their methodology also indicates that it is better to have the training component in free-living conditions. Other free-living studies have achieved better recognition accuracies but use simple classification problems, for instance Ellis et al. [10] had an average recognition accuracy of 85.6% but only distinguished sedentary, standing and walking movement, and driving. Likewise, Fullerton et al. [11] acquired 96.7% accuracy using broad activity label definitions (e.g., “Self-Care, “Home Activities”). The recognition accuracies in this investigation were superior to Liu et al [62] for a single accelerometer sensor (69.9%), comparable when multiple accelerometer sensors were used (74%), and inferior when they included an additional ventilation sensor (84.7%). This further reinforces that the performance in this investigation could have been improved with the additional sensors. In addition to neural networks, recognition of a wide variety of activities may be achievable through probabilistic models such as the General Probabilistic model proposed in [63] that build upon the recognition of simple motions (atomic activities) to recognize more dynamic and complex physical activities. Within the context of this pilot study, a probabilistic model could theoretically first recognize the condition of terrain (flat, hill or stairs) followed by the type of terrain traversed to build a complete activity label. Such models and are worth investigating in future studies.

Generally, there are not many studies which have attempted to differentiate walking across different terrains using only IMU data collected by humans. The work carried out by Hu et al. [64] appears to be the first study to attempt terrain classification with a single IMU, however this was performed in laboratory conditions and only distinguished between flat and uneven ground. Hashmi et al. [65] was able to acquire very high classification accuracies (87.5% with an SVM classifier) when distinguishing six types of terrain: concrete, tiles, carpet, asphalt, soil, and grass. Likewise, they also found consolidating terrains into hard and soft categories improved the average recognition accuracy to 92.08%, they however only tested flat terrain and did not include hills or stairs in the classification problem. As previously discussed in the methodology, the angle of slopes calculated during the study could not be validated, which may have had an impact on the accuracy of the ground truth, and so will require validation in a future study prior to collection activity data from participants. More recently, Russell et al. [66] carried out HAR in a cross-country trail. While the study acknowledged the presence of terrain, as well as the speed at which the terrain was traversed, their classification problem was simplified to only distinguish between laying, sitting, walking, running, or climbing a fence. Publication bias may explain the apparent absence of these types of studies: experiments which have attempted similar methodology to this investigation have achieved similar or worse results, and so have elected not to publish results.

The poor cross-subject validation results in experiment 3 reflect the findings of Obrien et al. [67], who tried training data on healthy populations and testing them on individuals who had suffered a stroke. In Obrien et al.’s findings, [67], the recall rate of classification accuracy was 53%, increased to 75% when trained on patients with only mild gait impairments. They also had the benefit of a significantly higher training population (15 healthy participants and 30 stroke patients), so from their results it is inconclusive whether it was the larger training dataset or the condition of the gait that was the key factor in improving performance. Due to only having four participants, this investigation did not have the capability of being able to investigate different types of severity in gait impairment for the ILLA population, and all four of the participants were considered to only to have mild gait impairment. There should also be consideration of the volunteering bias in this study: participants who are willing to volunteer will likely have a healthy lifestyle which will allow them to participate with minimal physical challenge, and the machine learning performance generated from this study may not reflect the performance if applied to the general ILLA population. Though the sample size was small enough that no comment can be made on statistical significance, the majority of recruited amputee participants had traumatic-related amputation, and traumatic amputees typically tend to exhibit greater physical and functional mobility over amputees with a vascular-related amputation [68]. Therefore, this study cannot implicate how effective the system would be on individuals who for instance have severe comorbidities impacting on their gait mobility. In contrast, Vageskar [69] was able to achieve much higher recognition accuracies when training on a healthy population, gaining 86.5% accuracy when trained in free-living conditions, however they were able to do so by simplifying the classification problem; most notably, they relabelled stair movement as just “walking” and did not account for sloping movement at all. Including those activities as separate labels would likely have caused a significant drop in their classification performance.

## 5. Conclusions

This pilot study was an investigation of how supervised learning systems can recognize activities carried out by healthy individuals and those with lower limb amputation, finding reasonable machine learning accuracies with room for improvement. SVM and LSTM classifiers were found to be the best off-the-shelf approach to classification of activities, achieving 77% recognition accuracies at a basic level of label resolution. Generally, there was scope to expand walking activities to include hard and soft terrains alongside walking uphill, downhill, upstairs, and downstairs. It was not feasible to try and recognize ILLA activities when the system was trained only on healthy individuals or other ILLAs, though a small recruitment sample and confounding factors on the types of terrains traversed by participants limit the generalizability of these findings. Future endeavours should investigate more intricate classifier designs and include a wider range of ILLA participants with varying degrees of amputation and mobility.

## Figures and Tables

**Figure 1 sensors-21-08377-f001:**
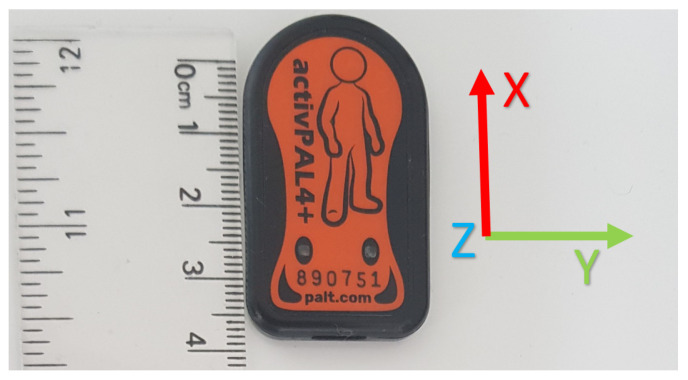
Scaled diagram of an ActivPAL device, with axes of orientation for reference.

**Figure 2 sensors-21-08377-f002:**
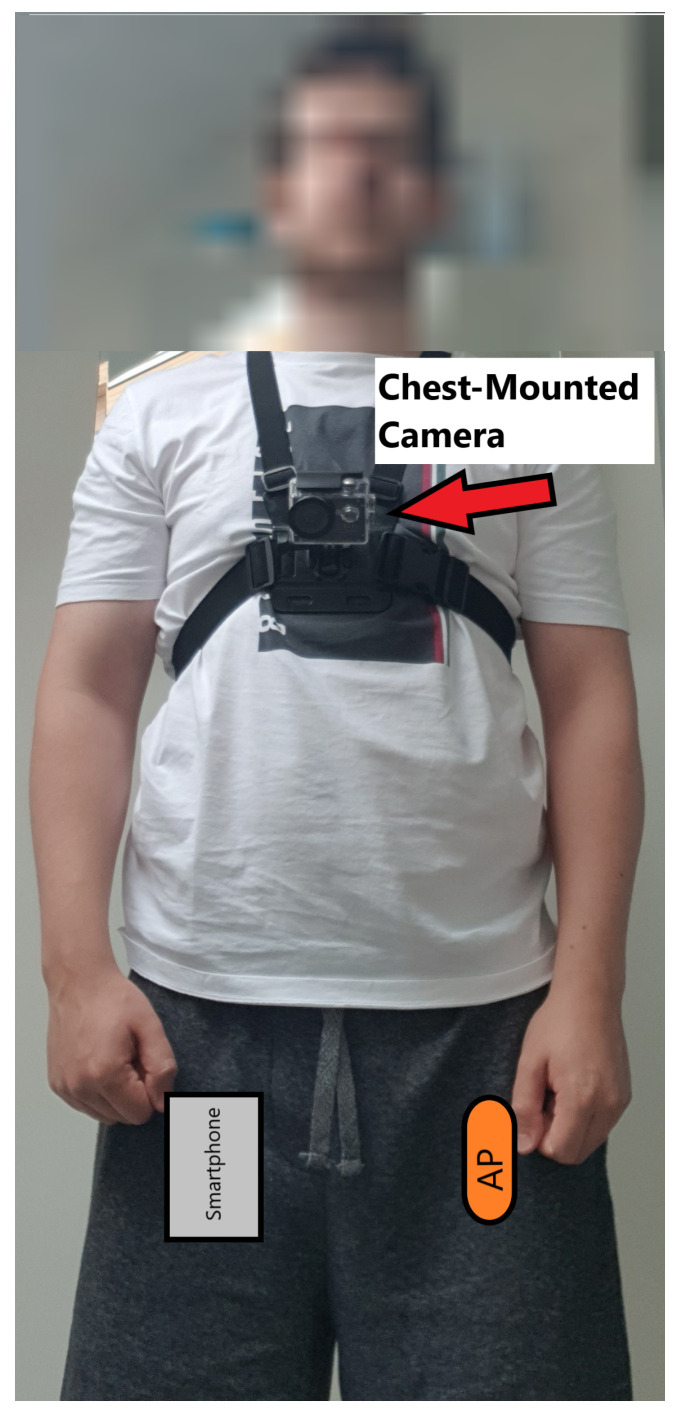
Example of a recording set-up. AP = ActivPAL. The smartphone and ActivPAL would both be worn under clothing.

**Figure 3 sensors-21-08377-f003:**
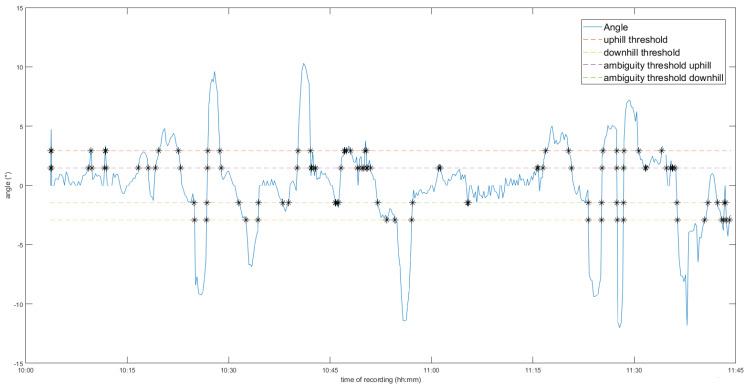
Example of a slope angle plot for one participant’s recording.

**Figure 4 sensors-21-08377-f004:**
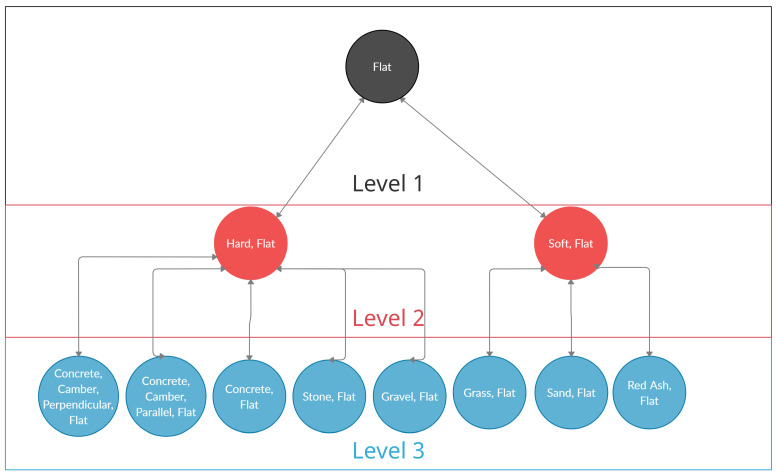
Example of the label consolidation process for “flat” activity labels.

**Figure 5 sensors-21-08377-f005:**
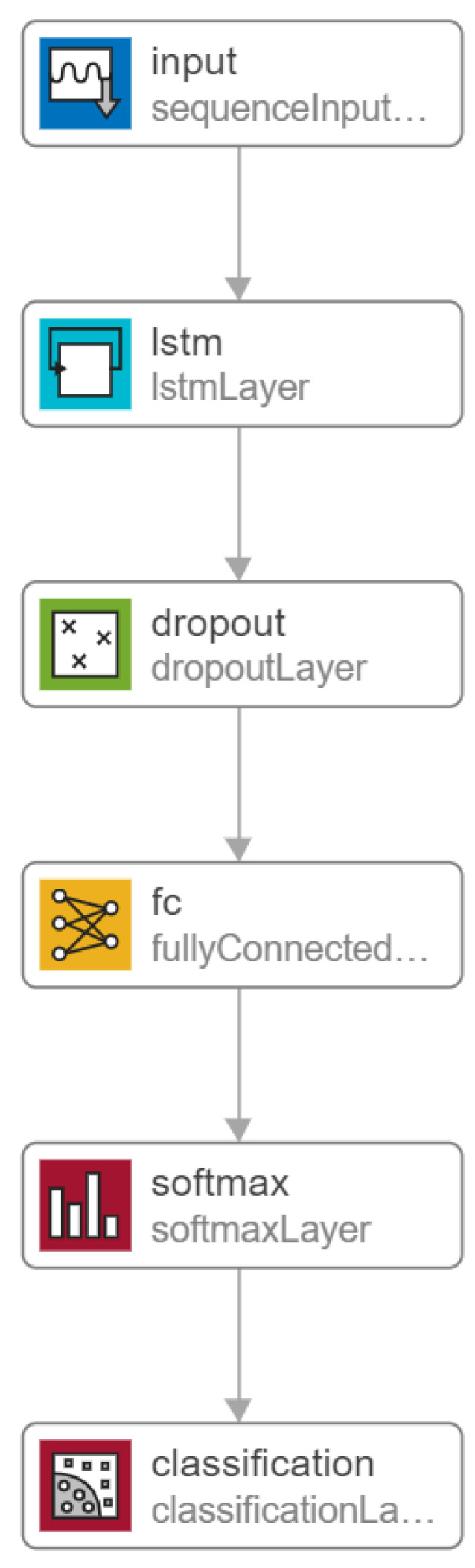
Overview of Long-Short Term Memory (LSTM) Matlab architecture. The input represents a 40 sample segment of a 3 dimensional (triaxial) input of raw accelerometer data.

**Table 1 sensors-21-08377-t001:** List of all collected activities in investigation and their total corresponding sample counts.

Terrain Label	Total No. of Samples
Concrete, Flat	12,200
Concrete, Camber, Parallel	3923
Concrete, Downhill	2549
Concrete, Uphill	2427
Grass, Flat	1346
Concrete, Camber, Perpendicular	877
Upstairs	691
Downstairs	656
Concrete, Camber, Downhill, Parallel	572
Concrete, Camber, Uphill, Parallel	540
Stone, Flat	496
Sand, Flat	232
Stone, Uphill	176
Grass, Downhill	151
Concrete, Camber, Downhill, Perpendicular	93
Concrete, Camber, Uphill, Perpendicular	93
Stone, Downhill	80
Grass, Uphill	78
Gravel, Downhill	46
Gravel, Flat	32
Sand, Uphill	31
Red Ash, Flat	26
Sand, Downhill	12
Gravel, Uphill	3

**Table 2 sensors-21-08377-t002:** Parameters tuned in Experiment 1.

Classifier	Parameters Tuned
SVM	Kernel Type, Box Size
KNN	Number of neighbours, Distance Metric, Distance Weight
Random Forest	Maximum number of learners, maximum number of splits
AdaBoost	Maximum number of learners, maximum number of splits, learning rate
Naïve-Bayes	Type of distribution, Type of kernel (for kernel distribution)
Discriminant Analysis	Type of discriminant analysis (Linear, Quadratic)
LSTM	Dropout factor, Number of Hidden Units

SVM—Support Vector Machine | KNN—k-Nearest Neighbour | LSTM—Long-Short Term Memory (Network).

**Table 3 sensors-21-08377-t003:** Characteristics of healthy participants without gait imapirment.

Participant	Height (m)	Weight (kg)	Age (Years)	Gender
#1	1.80	84	24	Male
#2	1.65	63	51	Female
#3	1.62	65	18	Female
#4	1.97	99	25	Male
#5	1.92	102	25	Male
#6	1.83	89	24	Male
#7	1.84	88	25	Male
#8	1.78	98	25	Male

**Table 4 sensors-21-08377-t004:** Characteristics of lower-limb amputated participants.

Participant	Height (m)	Weight (kg)	Age (years)	Gender	Type of Amputation	Type of Prosthesis	Origin	Years since Amputation
#1	1.79	95	55	Male	Unilateral transtibial	TSB socket, ESR foot	Traumatic	3
#2	1.70	86	57	Male	Unilateral transtibial	TSB socket, ESR foot	Traumatic	32
#3	1.72	110	40	Male	Unilateral transtibial	PTB socket, ESR foot	Traumatic	33
#4	1.52	61	48	Female	Bilateral transtibial	TSB socket, ESR foot	Vascular	4

TSB—Total Surface Bearing | ESR—Energy-Storing-and-Returning| PTB—Patellar Tendon Bearing.

**Table 5 sensors-21-08377-t005:** Comparisons of Supervised Classifiers under 5-fold Cross Validation and the best suited hyperparameters for each.

Classifier	5-Fold Accuracy (%)	Fold Std. (±%)	Best Parameters
SVM	77.22	0.54	Kernel: Gaussian | Box Constraint: 193
KNN	75.76	2.26	Num. Neighbours: 12 | Distance Metric: Correlation | Distance Weight: Squared Inverse
Random Forest	71.84	0.69	Num. Learners: 188 | Num. Splits: 4392
Adaboost	69.23	3.18	Num. Learners: 211 | Num. Splits: 910 | Learn Rate: 0.07
NB	64.40	0.95	Gaussian Kernel Distribution
LDA	73.91	0.93	Quadratic
LSTM	78.46	2.89	Dropout Factor: 0.2 | Num. Hidden Units: 190

SVM—Support Vector Machine | KNN—k-Nearest Neighbour | NB—Naïve Bayes | LDA—Linear Discriminant Analysis | LSTM—Long-Short Term Memory (Network).

**Table 6 sensors-21-08377-t006:** Comparison of mean classification accuracy in SVM and LSTM classifiers for varying resolutions of activity label, using 5-fold cross-validation.

Label Resolution Level	1	2	3
**SVM accuracy (%)**	77.22 ± 0.54	62.60 ± 2.73	32.74 ± 2.46
**LSTM accuracy (%)**	78.46 ± 2.89	73.77 ± 1.83	41.85 ± 4.19

SVM—Support Vector Machine | LSTM—Long-Short Term Memory (Network).

**Table 7 sensors-21-08377-t007:** Mean F1 Scores for Level 1 Label Resolution.

Activity	No. of Test Samples	LSTM F1 Scores	SVM F1 Scores
Downstairs	131	80.0	74.6
Upstairs	138	80.1	75.7
Flat	3826	83.8	84.2
Uphill	670	68.7	69.9
Downhill	701	62.8	61.7

SVM—Support Vector Machine | LSTM—Long-Short Term Memory (Network).

**Table 8 sensors-21-08377-t008:** Mean F1 Scores for Level 2 Label Resolution.

Activity	No. of Test Samples	LSTM F1 Scores	SVM F1 Scores
Downstairs	131	80.3	75.8
Upstairs	138	77.4	75.1
Hard, Flat	3506	80.2	82.3
Hard, Uphill	647	69.9	70.8
Hard, Downhill	668	63.1	61.5
Soft, Flat	321	57.1	61.3
Soft, Uphill	22	27.1	36.2
Soft, Downhill	33	27.6	36.7

SVM—Support Vector Machine | LSTM—Long-Short Term Memory (Network).

**Table 9 sensors-21-08377-t009:** Mean F1 Scores for Level 3 Label Resolution.

Activity	No. of Test Samples	LSTM F1 Scores	SVM F1 Scores
Concrete, Camber, Downhill, Parallel	114	34.8	38.0
Concrete, Camber, Downhill, Perpendicular	19	3.7	0.0
Concrete, Camber, Parallel	784	50.8	62.3
Concrete, Camber, Perpendicular	176	8.9	4.4
Concrete, Camber, Uphill, Parallel	108	39.3	50.5
Concrete, Camber, Uphill, Perpendicular	18	3.2	0.0
Concrete, Downhill	510	48.1	58.3
Concrete, Flat	2440	42.4	72.5
Concrete, Uphill	485	55.5	69.5
Downstairs	132	71.5	74.9
Grass, Downhill	30	31.5	37.1
Grass, Flat	269	51.0	62.6
Grass, Uphill	16	25.1	29.1
Gravel, Downhill	9	17.8	24.2
Gravel, Flat	6	7.4	0.0
Gravel, Uphill	1	0.0	0.0
Red Ash, Flat	5	17.4	25.0
Sand, Downhill	2	0.0	0.0
Sand, Flat	47	33.3	52.0
Sand, Uphill	6	18.6	27.0
Stone, Downhill	16	15.9	9.4
Stone, Flat	99	18.4	19.4
Stone, Uphill	35	38.3	41.3
Upstairs	139	69.4	73.4

SVM—Support Vector Machine | LSTM—Long-Short Term Memory (Network).

**Table 10 sensors-21-08377-t010:** Mean Leave-One-Subject-Out (LOSO) accuracies of each of the four lower limb-amputated subjects, using models trained only on individuals without gait impairment.

Participant	A1	A2	A3	A4	Mean	Inter-Subject Std.
**SVM accuracy (%)**	52.41	60.92	46.44	58.41	54.55	±5.61
**LSTM accuracy (%)**	24.97	48.71	14.79	25.60	28.52	±12.42

SVM—Support Vector Machine | LSTM—Long-Short Term Memory (Network).

**Table 11 sensors-21-08377-t011:** Mean Leave-One-Subject-Out (LOSO) accuracies of each of the four lower limb-amputated subjects, using models trained only on lower limb amputated individuals.

Participant	A1	A2	A3	A4	Mean	Inter-Subject Std.
**SVM accuracy (%)**	58.12	52.99	54.08	61.55	56.68	±3.40
**LSTM accuracy (%)**	53.97	10.26	43.22	16.96	31.10	±18.06

SVM—Support Vector Machine | LSTM—Long-Short Term Memory (Network).

## Data Availability

The accelerometer data presented in this study are openly available in Alexander Jamieson’s PURE account, DOI: 10.15129/ae451315-5258-4a07-8eb4-204e4d2e357f, see Jamieson [70]. The video files of the original recordings are not included in this dataset to protect the privacy of the participants.

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
