# Peer review of "Human Activity Recognition of Individuals with Lower Limb Amputation in Free-Living Conditions: A Pilot Study"

_sensors, 2021, doi:10.3390/s21248377_

Round 1

Reviewer 1 Report

The authors present a human activity based-IMU recognition on healthy subjects and amputees. The proposal included a variety of outdoor free-living activities and terrains in uncontrolled conditions and classification was categorized on three levels, from the simpler to the more complex, in the number of classes that composed the experiments. Is evident the high degree of difficulty of this problem, considering the variety of tasks and the movement task variability between lower-limb amputees and able-bodied subjects. The authors are aware of the limited number of participants, especially in the amputee group, which makes it difficult to obtain significant results. Although the unbalance data are a problem for pattern recognition problems, that reveals the difference of most common activities for both groups. The method used was completely described and followed meticulous statistical assessments.  The results are comparable with other cited studies. However, the discussion covers the difficulty of fairly comparing the results with previous studies due to the heterogeneity of the applications. Even, considering the above lack of this research, the approach has the potential to contribute to the proposed problem.

Next, some concerns are highlighted in order to improve the manuscript. 

The objective in the abstract (lines 2-4) as well as the introduction (lines 45-46) is unprecise and does not reflex precisely the work developed in the manuscript.

Methods:

The subjects collect data in only one day. Why it was not considered to increase the samples making other sessions?

Sampling frequency from measuring equipment, mainly the IMU sensor, is missing. In Line 86, it is mentioned the data segment in samples but sampling frequency is required.

Line 84. Include characteristics of the filter applied.

Line 173. It is mentioned that some parameters were excluded from the hyperparameter tunning, but they were not mentioned. 

Line 181. How was the selection of epochs? It was performed a random selection for data segments?

Discussion

Lines 310-326. It was unclear the purpose of the previous studies since it seems that they were not related to the current study. 

Minor concerns

Figure 3. Legend is too small for readers

Appendix B. The font size for Figures 1 from 5 is too small. 

Author Response

We thank you for your valuable comments. We revised the manuscript accordingly and in the attached file we present our response. All line, table, and figure numbers correspond to the revised manuscript.  

Reviewer 2 Report

The authors propose an approach of human activity recognition of individuals with lower limb amputation in free-living conditions. The topic of this paper is interesting. However, most content improvements are necessary.

1.Appears technically sound, although lack of clarity of explanation makes this difficult to judge in places. Parts of the methodology lack important detail, particularly regarding the classification models used for evaluation, upon which the results are based.

2.At present, many methods have been applied to human activity recognition. In addition to these methods adopted by the author, the author can also try other network structure models, which may improve the corresponding accuracy, e.g. LSTM variant model.

3.The tables needs to be unified.

4.The authors are suggested to find the following references for consideration:

Liu L, Wang S, Hu B, Wen JH, Qiong QY, Rosenblum, DS. Learning Structures of Interval-Based Bayesian networks in Probabilistic Generative Model for Human Complex Activity Recognition. Pattern Recognition, 81:545-561, 2018.

Whilst the proposed framework has some merit, this paper is let down by poor writing and significant omissions in methodology. Major revision recommended.

Author Response

We thank you for your valuable comments. We revised the manuscript accordingly and below we present our response to your comments. All line, table, and figure numbers correspond to the revised manuscript.  

Reviewer 3 Report

Attached.

Author Response

(The authors gave the same response as above.)

Reviewer 4 Report

This is a very good work. I believe it can be accepted for publication as it is.

Author Response

Point 1: This is a very good work. I believe it can be accepted for publication as it is.

Response 1: No further actions required based on comment.

Round 2

Reviewer 3 Report

The authors have addressed all of my concerns. However, I repeat my suggestion that "pilot study" should appear in the title.

Author Response

Title has now been modified to "Human Activity Recognition Of Individuals With Lower Limb Amputation In Free-Living Conditions: A Pilot Study"